# Crustal rheology controls on the Tibetan plateau formation during India-Asia convergence

Lin Chen[1], Fabio A. Capitanio[2], Lijun Liu[3] & Taras V. Gerya[4]

The formation of the Tibetan plateau during the India-Asia collision remains an outstanding issue. Proposed models mostly focus on the different styles of Tibetan crustal deformation, yet these do not readily explain the observed variation of deformation and deep structures along the collisional zone. Here we use three-dimensional numerical models to evaluate the effects of crustal rheology on the formation of the Himalayan-Tibetan orogenic system. During convergence, a weaker Asian crust allows strain far north within the upper plate, where a wide continental plateau forms behind the orogeny. In contrast, a stronger Asian crust suppresses the plateau formation, while the orogeny accommodates most of the shortening. The stronger Asian lithosphere is also forced beneath the Indian lithosphere, forming a reversed-polarity underthrusting. Our results demonstrate that the observed variations in lithosphere deformation and structures along the India-Asia collision zone are primarily controlled by the strength heterogeneity of the Asian continental crust.

[1] State Key Laboratory of Lithospheric Evolution, Institute of Geology and Geophysics, Chinese Academy of Sciences, Beijing 100029, China. [2] School of Earth, Atmosphere and Environment, Monash University, Clayton, Victoria 3800, Australia. [3] Department of Geology, University of Illinois at Urbana-Champaign, Champaign, Illinois 61820, USA. [4] Department of Earth Sciences, ETH-Zurich, Sonneggstrasse 5, Zurich 8092, Switzerland. Correspondence and requests for materials should be addressed to L.C. (email: chenlin@mail.iggcas.ac.cn).

The Tibetan plateau is one of the most striking features of continental tectonics, developed following the Indian continental subduction and collision with Asian continent. The convergence of the two plates, since ∼50 Ma, is continuously accommodated by Indian lithosphere subduction with its crust accreted into the Himalayan orogen, and by the indentation onto the Asian plate[1,2]. This process has presumably driven the northward thickening of the Tibetan crust (Fig. 1a). But how the Tibetan plateau formed and grew farther north from the Himalayan chain remains an outstanding question.

Several distinct models have been proposed to explain the growth of Tibet, including plateau formation as the consequence of the Indian crust underplating[3], distributed north-south shortening of Asian crust[4], lateral extrusion of crustal blocks along strike-slip faults[5], and channelized lower crustal flow[6,7]. The debate on the plateau formation mechanisms is further reflected in the complex seismic and magnetotelluric structures found beneath the Tibetan plateau along the whole convergent margin[8–12]. It is well established that the Indian lithosphere underthrusts the entire Himalaya and the southern portion of Tibet[8,10,13,14]. More to the north, the Indian lithosphere plunges into the mantle with a systematic increase in dip angle from west to east (Fig. 1b-d), as revealed by multiple recent seismic studies[10,15–17], where it subducts into the mantle. The convergent margin architecture becomes more complex towards the west, where the subduction polarity reverses beneath Pamir, with the Asian lithosphere becoming the down-going plate[16,17] (Fig. 1b). Similar vergence is found beneath the northern Tibetan plateau, where seismic imaging shows that the underthrusting Asian lithosphere under central Tibet[15] (Fig. 1c) becomes southward subducting further east[18,19] (Fig. 1d).

The seismic studies clearly suggest strong heterogeneities of the lithospheric structure along the entire collision zone, which likely inherited from the tectonic history of the Asian margin. The proto-southern margin of Asia was a collage of compositionally variable materials accreted onto the stable North Asian Siberian-Mongolian craton in the course of pre-Cenozoic orogenic events[1,20,21]. Thus, the net lithospheric strength must have varied largely along this margin[22]. Some of these heterogeneities might be preserved in the present-day structure of the lithosphere, such as the Tarim Basin, which is a rigid crustal block compared to the surrounding crust[23], and the inferred weaker crust in the southern and central Tibet[24–26]. Other lithospheric mantle heterogeneities have been inferred, such as through delamination or convective removal[27,28], which were likely achieved during the late stage of the plateau evolution. The crustal heterogeneities are likely representative of the strength of the Asian margin before Cenozoic collision. The inferred lateral variation of crustal strength strongly correlates with the present-day deformation, with a narrow plateau adjacent to the Tarim Basin and a broader plateau to the east (Fig. 1a). This suggests that east-west variations in the strength of the Asian plates might have played a dominant role for the development of the observed heterogeneous structure of the whole Himalayan-Tibetan system.

This study uses three-dimensional continental collision models to investigate the role of crustal heterogeneity in forming the orogen-parallel variations in lithosphere deformation and structures. We demonstrate that a weaker upper plate favors the formation of a wide continental plateau and two-sided subduction, while a stronger upper plate favors the development of a narrow orogeny and reversed-polarity underthrusting. Our models suggest that the lateral variations observed in the lithosphere deformation and structures along the India-Asia collision zone are controlled by the strength heterogeneity of the Asian continental crust.

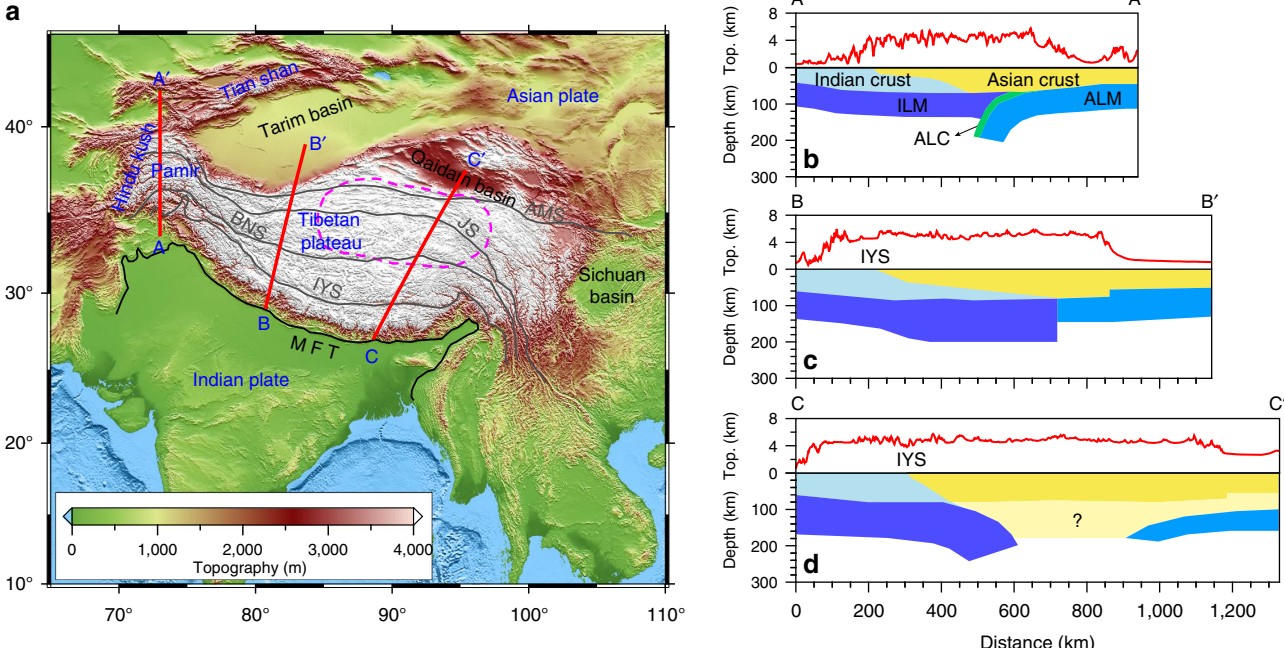

**Figure 1 | Surface topography and lithospheric structure within and around the Tibetan plateau. (a)** Topographic map around the Tibetan plateau. Red lines indicate the locations where lithospheric structures are shown on the right. Black and grey lines indicate the sutures and main frontal thrust (MFT), respectively. IYS, Indus-Yarlung suture zone. BNS, Bangong-Nujiang suture zone. JS, Jinsha suture zone. AMS, Anyimaqen-Muztagh suture zone. The magenta dashed curve outlines the zone of weak Sn signal[13]. **(b-d)** Lithospheric structural configuration beneath profiles AA'[16,17], BB'[15] and CC'[18,19], respectively. ILM, Indian lithosphere mantle. ALM, Asian lithosphere mantle. ALC, Asian lower continental crust. The question mark in **d** indicates the seismically unresolved region, where the weak $S_n$ signal was found[13] (see magenta dashed line in Fig. 1a).

## Results

**Model setup**. We develop three-dimensional thermo-mechanical numerical models with realistic lithosphere rheologies (Methods, Supplementary Fig. 1) to provide insight on the evolution of coupled subduction and collision along the India-Asia convergent margin. We explore three different sets of models to understand the role of a strong and weak lower crust in the Asian continent during convergence and indentation (Supplementary Fig. 1 and Supplementary Tables 1 and 2). We use flow laws of 'dry Diabase'[29] and 'Mafic Granulite'[30] to represent the stronger lower crust of Tarim Basin and the weaker lower crust of the rest Tibet, respectively. Here, we present two end-member models embedding a weak or strong lower crust, and a third model embedding both types of lower crust juxtaposed along the convergent margin. The investigated lower crustal strength varies up to ca. three orders of magnitude and provides a range of realistic lithospheric strengths (Supplementary Fig. 2). An assessment of lower crust and lithospheric mantle rheologies demonstrates that the lower crust heterogeneities have first-order control on the whole lithosphere strength. Northward convergence velocities are imposed at the first half of the front wall with rates of 2.0, 3.3 and 5.0 cm per year, spanning the range of Cenozoic southern Asian convergence rates[19]. We take the case with 3.3 cm per year convergence rate, which is in agreement with the average Cenozoic Asian shortening rate[31], as the reference model. In the remainder of the paper, we refer to the incoming plate as the Indian plate or indenter, and the broader retro-continent as the Asian plate or upper plate.

**Upper plate's strength controlling deformation style**. Modelling results show critical dependence of the collision style, lithospheric structures and surface morphology on the strength of the lower crust (Figs 2 and 3). A weak upper plate in the retro-land of the plate margin accommodates part of the collision, by thickening and back-thrusting, resulting in a coupled evolution of the orogen and the plateau in the continent interiors, which broadens with increasing convergence (Fig. 2a). Instead, a strong upper plate acts as a rigid buttress to the orogeny and collision is accommodated along the margin in a narrow orogen, as the rigid upper plate resists strain propagation (Fig. 2b). When the strength of the upper plate varies along the margin, combining the two end members in a single model, the orogeny-plateau system develops strong heterogeneities with both styles represented (Fig. 3).

**Weak Asian crust favouring a wide plateau**. In the weak Asian lower crust model (Fig. 2a), the India-Asia plate interface migrates progressively northward accompanying Indian subduction. An orogenic wedge develops above the down-going plate, and large-scale crustal thickening (Fig. 4a) and a broad plateau forms in the interior of the upper plate (Supplementary Fig. 3). The interface between the two plates reaches as deep as ∼150–200 km, for the range of convergence rates tested. The strong down-going plate drags along the upper plate, while most of the crusts are scraped off, forming an orogenic wedge above. The boundary between the two plates at the surface, referred to as suture zone (Fig. 2a), is found ∼70–100 km towards the foreland from the deep plate interface. The orogen develops mostly within the upper crust, as a result of detached lower crust along a frontal thrust. Large deformation is accommodated in the upper plate's crust by deep-seated back-thrusting (the southward yellow-faced extrusion in Fig. 2) and diffuse thickening, leading to the formation of a very broad plateau behind the orogen. The crust in the upper plate's interior thickens up to ∼90 km (Fig. 2a). Both flexural and isostatic responses of the whole lithosphere control

the uplift of the plateau, whose area is wider than that of the thickened crust (Fig. 4a).

**Strong Asian crust favouring a narrow orogen**. In the model where the Asian plate's lower crust is stiffer than that of the Indian plate, a narrow orogenic wedge develops above the convergent margin and plateau growth is inhibited. In this model (Fig. 2b and Supplementary Fig. 4), crustal deformation is accommodated along a narrow orogenic wedge, where mostly the Asian crust is accreted and thickened (Fig. 4b). The orogenic wedge eventually grows to a height of 7–8 km, but remains narrower than ∼150 km, without plateau development further inland. The crustal evolution shows little propagating deformation into the Asian plate in this model, and major crustal thickening, up to ∼90 km, is found only in a narrow zone, which does not widen during convergence (Fig. 4b). This result is largely insensitive to the different convergence rates considered and, although slower convergence favors strain diffusion, no plateau forms behind the narrow orogen. Interestingly, this model develops a different deep lithospheric structure from the case with a weaker crust, in that the Asian lithosphere underthrusts beneath the Indian margin (Fig. 2b). This suggests that the stiffer Asian crust affects the strength of the entire lithosphere, which resists internal deformation and forces its way under the indenter. Upon ongoing convergence, the coupled lithosphere and lower crust reaches mantle depth, eventually evolving into a reversed polarity of subduction (Fig. 2b). In contrast, the lithosphere with a weaker Asian crust tends to become the overriding plate, where more internal deformation occurs in both the crust and underlying mantle (Fig. 2a).

**Asian crustal heterogeneity shaping the orogenic plateau**. When both a stiffer and a weaker lower crust are embedded in the model's Asian plate from west to east along the convergent margin, the orogen-plateau system develops both styles (Fig. 3 and Supplementary Fig. 5). A narrow orogen forms above a southward subduction to the west, and a wider plateau further develops behind the orogen to the east, where the subduction polarity switches to northward. In this case, the resulting topography and crustal thickness (Figs 3 and 4c) are similar to the two end members (Figs 2a,b and 4a,b), showing that the initial configuration of crustal strength controls how the orogen-plateau system evolves.

## Discussion

These models offer relevant insights in the complexities found along the India-Asian margin and the formation of the coupled orogen and plateau as the result of varying crustal rheologies. The Himalayan orogen is bounded by a deep-seated thrust fault, the Main Himalayan Thrust, revealed by seismic investigations in the southern Tibet[32–34]. This fault functions as a crust–mantle decollement, and controls the growth of the crustal-stacking wedge[35]. The exposure of the two plates' crustal interface at the surface, corresponding to the Indus-Yarlong suture (IYS) (Fig. 1a), is found ∼100 km south of the northern limit of the Indian plate. The presence of such an offset is consistent with the seismically imaged northern limit of the Indian lithosphere underplating further north of the IYS[33,36]. Models with a weak Asian crust, or portions of it, reproduce structures compatible with the observations beneath the Himalaya-Tibet system, where similarly weak crust has been inferred[37]. These models reproduce the observed main features such as the basal décollement, the down-going lithosphere underplating and the location of the suture above the subducting plate, as well as a wide plateau in the upper plate interior (Figs 2a and 3).

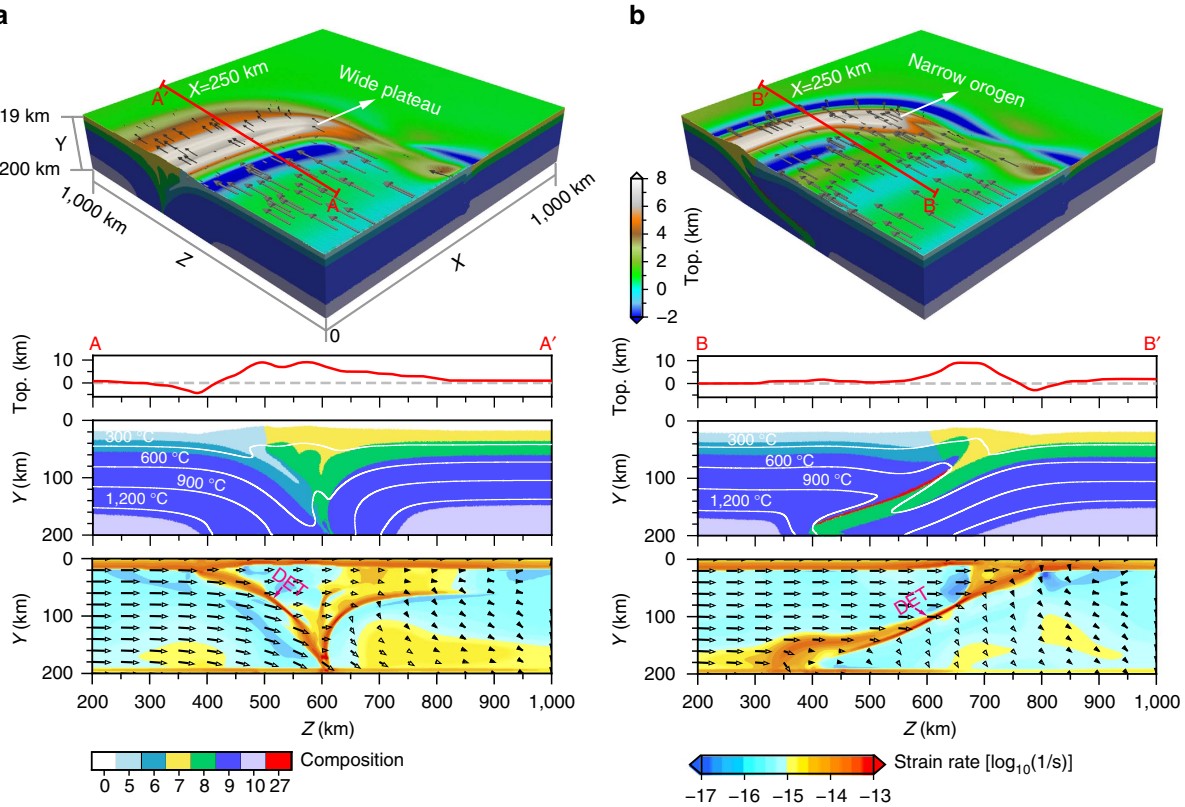

**Figure 2 | Snapshots of continental indentation.** The displayed models are (**a**) 'Model-1' (weaker Asian crust model) and (**b**) 'Model-2' (stiffer Asian crust model) at 17.7 Myr, corresponding to a total convergence of ∼580 km; see Supplementary Table 1 for more model details. In **a**, a broad orogenic plateau forms; in **b**, only a narrow orogenic wedge forms. (upper, middle and lower) Three-dimensional tectonic deformation with topography and surface velocities, cross-sections of composition field with topographic profiles over plotted and isotherms overlapped, and cross-sections of the second strain rate invariant (arrows show resulting velocities), respectively. Red lines in the top panel indicate positions of the two-dimensional cross-sections shown below. Colour codes for composition and strain rate are shown at the bottom of the figure, and that for topography is shown in the middle of the upper panel. For composition codes, 0-sticky air; 5-upper continental crust (indenter); 6-lower continental crust (indenter); 7-upper continental crust (upper plate); 8-lower continental crust (upper plate); 9-lithospheric mantle; 10-asthenosphere; 27-partially molten continental crust. DET, crustal-scale detachment faulting.

Along the western Tibet, the rigid Tarim Basin lithosphere might have acted as a rigid block resisting northward expansion of the plateau[38]. Observed lithospheric structures are mostly explained by the collision onto a stiffer crust, which enhances the back-thrusting and inhibits plateau growth, while forcing deep reversed-polarity underthrusting and subduction (Fig. 2b). Moho depth estimates in this area indicate a wide area with thicknesses in excess of 80 km beneath the southernmost portion of the Tibetan plateau, whereas similar thicknesses are found in a narrow belt beneath the western Tibet[2]. The crustal thicknesses and distribution beneath these two domains are compatible with the outcomes of the model where the two crustal rheologies are embedded (Fig. 3).

Furthermore, rheological heterogeneities in the crustal blocks in the Tibetan plateau interiors have possibly played a major role during the northward Cenozoic deformation propagation[21]. Crustal back-thrusting and southwards lithospheric under-thrusting beneath the northern Tibet and Qaidam Basin[15] developed along block boundaries where the nature and geological history, and hence rheology, differ largely[1]. South-verging underthrusting of the Asian lithosphere is seismically imaged beneath Pamir[16,17] (Fig.1b), with the lithosphere reaching mantle depth. A relatively stronger lower crust is inferred in the area stretching from northern Tibet to Pamir-Tien Shan[16], where the crust was likely thickened before collision[39]. Therefore, our models potentially explain the tectonic regime inversion along

the northern Tibetan to Pamir margin during its northward expansion, as the consequence of encountering a stronger Asian crustal block.

Further insight into the relation between the Tibetan plateau formation and the lower crustal rheology comes from the analyses of observed and modelled surface velocity and strain rates (Fig. 5). Global positioning system measurements show that the northward component of the velocity relative to Asia decreases steadily across the Tibetan plateau and almost vanishes at the northern margin[40]. To the east, the decrease in the convergence-parallel component is accompanied by an increase in the convergence-perpendicular component, exhibiting a clockwise rotation. To the west, the measurements indicate a reduction of the northward component across the relatively narrow plateau, which vanishes in the Tarim interior. The weaker crust model predicts a steady northward decay of the surface velocities across the broad plateau (Fig. 5a), in agreement with the global positioning system pattern of the eastern Tibet, whereas the stiffer crust model predicts no substantial reduction in the surface velocities across the narrow wedge but abruptly vanishing in the stiffer crust block north of the indenter (Fig. 5b), resembling the observations in the western Tibet adjacent to the Tarim Basin. More importantly, our models provide insights in the ongoing deformation in the area. The high strain rates predicted by the third model, embedding a stiffer and a weaker crust, localize along the frontal thrusts of the orogen, the southern and northern margin of the plateau, although smaller and more

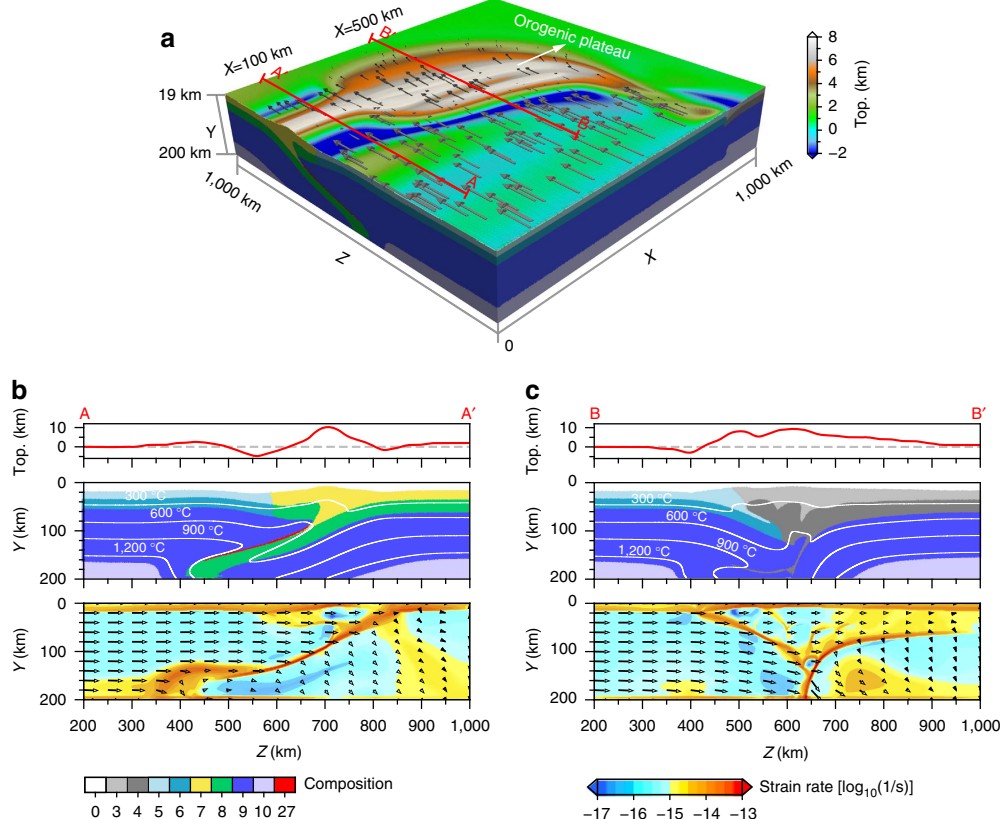

**Figure 3 | Combined model embedding a stiffer and weaker lower crust.** The displayed model is 'Model-3' (the third model) at 18.0 Myr; see Supplementary Table 1 for more details. (**a**) Snapshot showing three-dimensional model deformation with topography and surface velocities at the model surface. Cross-sections in **b,c** show the composition field with topographic profiles over plotted and isotherms overlapped and the second strain rate invariant with modelled velocities overlapped along the profiles of $X = 100$ km and $X = 500$ km (as indicated by the red lines in **a**), respectively. Colour codes for composition and strain rate are shown at the bottom of the figure, and that for topography is shown at the right side of the three-dimensional snapshot. For composition codes, 0-sticky air; 3-upper continental crust (weaker crust block); 4-lower continental crust (stiffer crust block); 5-upper continental crust (indenter); 6-lower continental crust (indenter); 7-upper continental crust (stiffer crust block); 8-lower continental crust (stiffer crust block); 9-lithospheric mantle; 10-asthenosphere; 27-partially molten continental crust.

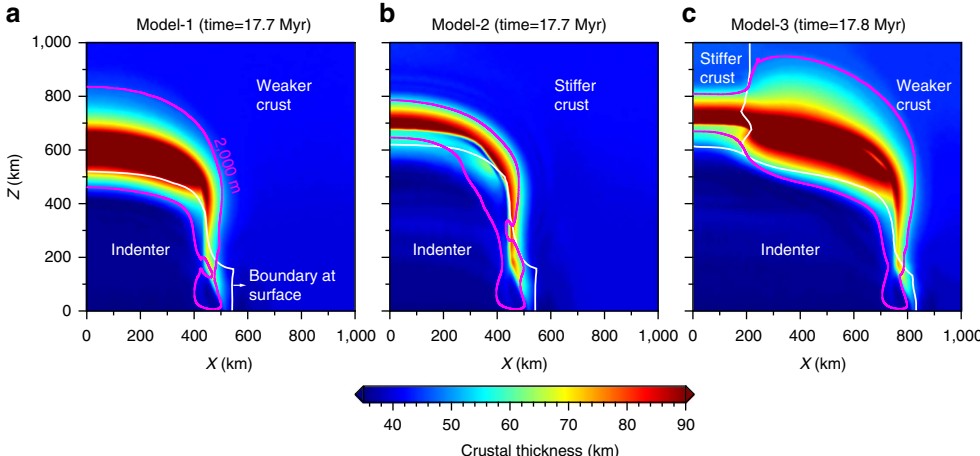

**Figure 4 | Modelled spatial distribution of crustal thickness.** (**a**) Crustal thickness distribution for the weaker crust model (Model-1; see Supplementary Table 1 for details.) at 17.7 Myr. (**b**) Crustal thickness distribution for the stiffer crust model (Model-2; see Supplementary Table 1 for details.) at 17.7 Myr. (**c**) Crustal thickness distribution for the combined crust model (Model-3; see Supplementary Table 1 for details.) at 17.8 Myr. Purple lines encompass the regions where the topography is above 2,000 m, and white lines mark the boundary between the different continental blocks at the surface.

diffuse in the north, while a relatively low strain rate zone is found in between (Fig. 5c). This spatial distribution is consistent with the mapping of the current strain rates within the Tibetan plateau, where inferred high strain rates on the southern and northern margins bound a relative low strain rate zone in the central plateau[41]. The strain rate pattern of the third model shows that the stiffer crust favors stress focusing and strain localization in the weaker domain. In the weaker crust portion, shearing along narrow

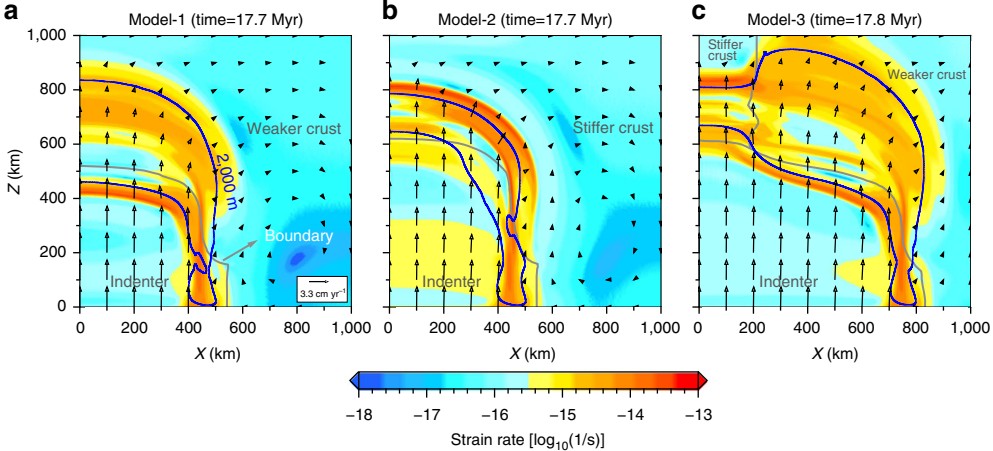

**Figure 5 | Modelled spatial distribution of the second strain rate invariant at the surface.** (**a**) Strain rate distribution at the surface for the weaker crust model (Model-1), showing diffuse deformation. (**b**) Strain rate distribution at the surface for the stiffer crust model (Model-2), showing localized deformation. (**c**) Strain rate distribution at the surface for the combined crust model (Model-3), showing localized deformation to the model west and diffuse deformation to the model east. Blue lines encompass the land above 2,000 m, and grey lines mark the boundary between the different continental blocks at the surface. Black arrows represent velocity vectors at the surface.

belts, likely thrusts, is promoted beneath the wide plateau, as opposed to the diffuse deformation in the weaker crust-only model. This might allow inferences on the deep lithospheric structure of the Tibetan Plateau, where similar narrow thrusts have been proposed[21], and the localization along narrow strain belt such as the Kunlun Fault[42], although this might be speculative.

Concluding, the presented models allow reconciling the formation of the Tibetan plateau with that of the Himalayan orogen, suggesting that they have a coupled evolution as the result of the relative strength of the interacting plates during indentation. Because similar heterogeneities in crustal types are common in continents, these must play a dominant role in the development of mountain chains and plateau topography around collisional margins.

## Methods

**Modelling approach.** The code I3ELVIS, which combines a finite-difference method and a marker-in-cell technique[43,44], is used for the modelling of continental indentation. It solves the three-dimensional momentum, mass and energy conservation equations on a staggered Eulerian grid system. Physical properties are transported through the motion of Lagrangian markers, which is driven by the velocity field interpolated from the Eulerian grid. The code considers non-Newtonian viscous-plastic rheologies for different lithologies (Supplementary Table 2), which is fully coupled with various thermal processes, including adiabatic, radioactive, latent and shear heating. Full details of the method, which allows for the reproduction, can be found in ref. 45.

**Numerical model design.** The initial model setup is built on a simplified tectonic map of the India-Asia collision zone (see Fig. 1a). The overall geometry of the model captures the configuration when the Indian continent started to indent into the Asian continent. The indenter (that is, Indian plate) is represented with the cyan-coloured region in Supplementary Fig. 1. The starting point of all simulations is assumed to be the beginning of the India-Asia hard collision. The presented simulations do not take into account the effects of the subduction of the Tethyan Ocean before the collision. The computational domain is equivalent to a box of $1,000 \times 200 \times 1,000$ km (in the order of $x$, $y$ and $z$), which is resolved by $501 \times 101 \times 501$ Eulerian nodes with a uniform resolution of $2 \times 2 \times 2$ km (Supplementary Fig. 1). There are more than 200 million Lagrangian markers, which are randomly distributed in the entire domain.

Free slip boundary condition is used for the back ($z = 1,000$ km), left ($x = 0$) and right walls ($x = 1,000$ km). A constant normal velocity ($\mathbf{v}_i$) is imposed at the first half (that is, $x = 0$–500 km) of the front wall, which simulates the northward convergence of the Indian lithosphere (see Supplementary Fig. 1), while free slip boundary condition is applied to the second half (that is, $x = 500$–1,000 km) of the front wall. To predict realistic topography, the free-surface boundary condition is implemented by adding a 'sticky air' layer with low density (1 kg m$^{-3}$) and reduced viscosity ($10^{18}$ Pa s) at the top of the model[46], which is 19 km thick above

the upper plate (that is, Asian continent) and 20 km thick above the indenter (that is, Indian continent). To keep mass balance within the 'sticky air' layer, an outflow velocity ($\mathbf{v}_a$) is applied at the model top, which is given by $\mathbf{v}_a = \left( \frac{W'_x}{W_x} \frac{t_a}{W_z} \right) \times \mathbf{v}_i$, where $t_a$ is the thickness of the 'sticky air'; $W_x$ and $W_z$ are the model width in the $x$- and $z$-direction, respectively; and $W'_x$ is the horizontal range on the front wall, where the convergence is imposed. An infinity-like free slip boundary condition is used for the lower boundary[47], which implies that the lower boundary is open for mantle flow.

The initial thermal structure is prescribed by linearly increasing the geotherm from 0 °C at the surface to 1,360 °C at the base of the lithosphere. The initial geotherm at the junction of two lithospheres with different thicknesses is calculated by linearly interpolating the isotherms of the neighbouring regions. An adiabatic thermal gradient of 0.5 °C km$^{-1}$ is used for the asthenospheric mantle. The thermal boundary conditions are 0 °C at the top and zero horizontal heat flux on all the vertical boundaries. An infinity-like external constant temperature condition is used for the lower boundary, which allows the temperature and heat flux at the lower boundary to be adjusted dynamically during simulation[45].

The reference model domain is composed of two continental lithospheres with different characters (Supplementary Fig. 1). The corner situated in the third quadrant ($0 \le x \le 500$ km and $0 \le z \le 100$ km, see Supplementary Fig. 1) is composed of a thicker/colder continental lithosphere with a crust at normal thickness, representing the cratonic Indian lithosphere. It initially has a 35-km-thick continental crust (a 15-km-thick upper crust and a 20-km-thick lower crust) and a 105-km-thick lithospheric mantle. The rest of the model domain initially has a 40-km-thick continental crust (a 17-km-thick upper crust and a 23-km-thick lower crust) and 80-km-thick lithospheric mantle, representing the proto-Asian lithosphere.

**Rheological model.** The strength of the lithosphere is evaluated, at the geological timescale, by the combination of brittle (plastic) and viscous rheology. The plastic rheology is implemented through the Drucker–Prager plasticity, which describes the linear dependence of the materials' resistance on the total pressure. It is calculated as[48]

$$\sigma_{\text{yield}} = C_0 + P\sin(\varphi_{\text{eff}}) \tag{1}$$

$$\sin(\varphi_{\text{eff}}) = \sin(\varphi)(1 - \lambda) \tag{2}$$

$$\eta_{\text{plastic}} = \frac{\sigma_{\text{yield}}}{2\dot{\varepsilon}_{II}} \tag{3}$$

where $\sigma_{\text{yield}}$ is the yield stress; $P$ is the dynamic pressure; $C_0$ is the cohesion at $P = 0$; $\varphi$ is the internal frictional angle; $\lambda$ is the pore fluid coefficient that controls the brittle strength of fluid-containing porous or fractured media; $\dot{\varepsilon}_{II} = \left( 0.5\dot{\varepsilon}_{ij}\dot{\varepsilon}_{ij} \right)^{1/2}$ is the second invariant of the strain rate tensor $\dot{\varepsilon}_{ij}$; $\eta_{\text{plastic}}$ is the viscosity for plastic rheology; and $\varphi_{\text{eff}}$ is the effective internal frictional angle that integrates the effects of internal frictional angle and pore fluid coefficient.

The viscous rheology is computed as[48]:

$$\eta_{\text{ductile}} = (\dot{\varepsilon}_{II})^{(1-n)/n} A_D^{-1/n} \exp\left( \frac{E_a + PV_a}{nRT} \right) \tag{4}$$

where $A_D$ (material constant), $E_a$ (activation energy), $V_a$ (activation volume), $n$ (stress exponent) are experimentally determined flow law parameters; $R$ is the gas constant; and $T$ is the absolute temperature.

The effective viscosity is calculated by comparing plastic and viscous rheologies as a function of depth[48]

$$\eta = \min\left\{\eta_{\text{plastic}}, \eta_{\text{ductile}}\right\} \qquad (5)$$

In the modelling, the experimentally determined flow law of 'Wet Quartzite' is used for the continental upper crust, 'Plagioclase $An_{75}$' for the Indian continental lower crust, and 'Dry Olivine' for the lithospheric and asthenospheric mantle[48]. The flow law used for the Asian continental lower crust is assumed to be either 'Mafic Granulite'[30] or 'Dry Diabase'[29], depending on experiments. The rheological and other parameters used here can be found in Supplementary Table 2.

**Data availability.** All the relevant data and model output presented in this study are available from the corresponding author upon reasonable request.

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

## Acknowledgements

We thank Xiaohui Yuan (GFZ) for comments on an earlier draft. We are grateful to two anonymous reviewers and Sheng Jin for their constructive comments. This study was supported by the Strategic Priority Research Program (B) of the Chinese Academy of Sciences (XDB18000000), by the National Key Research and Development Project (2016YFC0600406, 2016YFC0600101), and by the National Natural Science Foundation of China (41474084, 41490610). Open source software ParaView (http://www.paraview.org) was used for 3-D visualization.

## Author contributions

L.C. conceived the study, performed numerical experiments, interpreted results and wrote the manuscript. F.A.C. and L.L. contributed to the model tests and interpretation and wrote the manuscript. T.V.G. provided the 3-D thermo-mechanical code and guidance on using it and wrote the manuscript.

## Additional information

**Competing interests:** The authors declare no competing financial interests.

