## [Peer Review File · Nature Communications]

Reviewers' Comments:

Reviewer #1 (Remarks to the Author)

Review of "Crustal rheology controls on the Tibetan plateau formation during India-Asia convergence"

By Chen et al.

This paper deals with an outstanding question: what controls the development of the Tibetan plateau during India-Asia convergence. The paper is very well written and the methods used (numerical modelling) are very appropriate to tackle this objective. By varying the upper plate rheological layering (reflecting some tectonic inheritance in the natural case study), the authors argue that the development of a Plateau requires weak upper plate, in particular a relatively weak lower crust compare to the lower plate. Moreover, the strength/rheology of the upper plate also controls the subduction polarity.

I have found some weaknesses that need to be amended before possible publications.

In the introduction, the authors discuss in details tectonic inheritance within the Tibet Plateau, related to Cenozoic deformation for example. And these inherited structures are only simplified in this study by using differences in lower crust rheology. Could you please explain what justify this simplification? Why didn't you test different mantle rheology, or upper crust rheology? Many recent studies indeed argue that tectonic inheritance may imply strong variation in lithosphere mantle strength and therefore rheology.

The authors have selected very specific rheological parameters (Dry diabase or mafic granulite) for the lower crust. It is important that the authors discuss and justify their choice (among a lot of possible crustal rheological parameters), since it controls most of their results.

More important, I did not find in the text the mechanical explanation for such a strong effect of the upper plate lower crust rheology on the style of subduction/collision. Such a mechanical explanation is however necessary if you want to convince that you have find the process that explain the development of the Tibet Plateau.

Finally, you aim at modelling, in 3D, the Asia/India system, with effect of tectonic inheritance. A concluding model is to me missing, with the entire indenter (India), delimited in space to the East and West, and with variable (vertically and laterally) rheologies in the upper plate to better discusses the Tibet/Asia collisional system. I am suggesting this final model because it seems that, based on all the modelling material you provide, that you have the technical capacities to run it.

In conclusion, I found the idea of the paper very interesting but the authors need to amend the present manuscript in order to strengthen their arguments and conclusions. On these bases, I can only suggest MAJOR REVISION.

Reviewer #2 (Remarks to the Author)

In this paper, the authors use 3D numerical thermo-mechanical modelling to investigate the control of Asian crustal rheology on the development of the East-West heterogeneous structure of the whole Himalayan-Tibetan orogeny. They highlight that weak Asian crust will favour the formation of a plateau as part of the orogeny whereas strong Asian crust leads to the development of a narrower orogeny with no plateau.

The paper shows some interesting results on how crustal rheology has affected the orogen development into a plateau or not. Although the model seems to reconcile quite well with studies from the Pamir region, the results on the formation of the plateau in Tibet appear relatively minor compared to issues related to the growth of the plateau that have been discussed in many studies: extension of the Indian plate to the north, convective removal of the lithosphere, crustal thickness relation to topography, temperature evolution, the presence of partial melt in the crust, convective thinning of the lithosphere, For instance, the results are not really discussed in comparison with the proposed mechanisms involved in the growth of the Tibetan plateau presented in the introduction. The only discussed issues are mainly the significant tectonic feature of the Himalayan-Tibet orogeny appearing in the models as well as comparison with modelled and present day observed strain rates. Therefore, despite presenting interesting results on the orogeny formation, the paper remains quite vague into really answering the question it poses concerning the growth of the plateau. It is not clear if the authors aim at explaining the initiation of the plateau or its full development. The results remain very interesting in term of describing the initiation of the deformation but it feels it is not enough to answer the question about the formation of the plateau up to its present day state.

Please find below some comments and suggestions which I hope are clear enough and could help make the manuscript results more relevant.

Comment #1: The authors claim in their conclusion at line 174 that "the weak Asian crust can explain the formation of the plateau behind an orogeny wedge and the observed undeplating Indian lithosphere". However, from their results it is difficult to see how the actual observations from the state of the plateau today can be related to the models.

The paper would be a bit more clear maybe if Figure 2 would be replaced by Figure 3 to present the initiation and stages of the deformation first and then show on Figure 3 the final stage of the simulation to discuss how it compares with current topography and geophysical observations from Tibet and Pamir.

The topography should be compared with real topo data to discuss the differences, for instance, from the Figure 3 the plateau generated from model 1 seems to be only 200 km wide, how does this compare with real topography data which clearly reflects a wider plateau.

Also how does the thickness of the crust and temperature evolution through the models compare with the studies from the plateau?

Comment #2: The 3D code used is described as "Fully thermodynamically coupled and accounts for melting processes". A wide variety of studies have agreed on the presence of partial melt in the middle/lower crust of the plateau that would affect strongly the rheology of the plateau. From Figure 2, it seems that melt appears on model 2 but not model 1. It is worth discussing this point, why melt is not needed, how do both crustal thickness and geotherm evolve, how does this affect melt generation or not, and therefore the rheology. Maybe a closer focus on the crust in a picture could also help.

Comment #3: It is in some way paradoxical to discuss the heterogeneity of the whole orogen by comparing two "homogeneous" models with different rheology. It is worth to justify that choice. The authors use a 3D modelling so maybe it makes more sense to eventually include the crustal heterogeneities directly into one single model if computationally possible. The authors discuss the heterogeneities between Pamir and Tibet but they should also probably emphasis on the heterogeneities inside the plateau between west, central and eastern Tibet.

Comment #4: line 122 to 135, It is interesting to compare the model features with observed structures but those are very difficult to see on the Figures. The authors need to find a way to describe it on the models better. This comment goes back to a previous comment on having a picture of the model more focused on the crust.

Comment #5: line 144 to 150, it is actually a very important point that I think the authors should

have explored further particularly in the sense of trying to understand how the plateau extended to the north, by discussing the different mechanisms involved in the process as well as the rheological differences between west, central and east Tibet.

Comment #6: On the GPS comparison, the authors tend to repeat themselves and should maybe instead go more in detail in the comparison between synthetic and observed values of strain rates. On Figure 4, are the arrows velocities? Should be clarified. The results are compared with present day GPS measurements but what time of the simulation is represented on Figure 4. How does it compare then with real data? This section is relatively long and therefore looks like a major part of the manuscript. The authors should find a better way to compare those in the text and in the pictures.

Other comments

Line 40: On the descriptions of the models, the authors should also mention the effects of convective thinning of the lithosphere to the north which has been proposed as another mechanism to explain the extension of the high topography further north (See for example Jimenez-Munt 2008).

Line 44: Not only seismic but a variety of other studies such as magnetotellurics have contributed to study the deformation of the plateau.

Line 47: More recent references than Barazangi and Ni, 1982 have been published.

Line 49: In addition to Zhao et al. 2010, the authors could also add Agius and Lebedev, 2013 which also highlights quite well the heterogeneities of the lithospheric structure of the plateau and particularly discuss its effects on thermal anomalies. The later could be a very good addition to the temperature data extracted from the 3D simulations.

Line 52: (Fig. 1b) instead of (Fig. 1d)?

Line 51-54: No mentioning of Fig. 1c, maybe some short sentence would be nice.

Line 54: It should be discussed what is defined as an interrogation mark on the Fig. 1d.

Line 60: Maybe Yin and Harrison, 2001 could be a good reference here also as it is a review of the geology of Tibet.

Line 63: What about the strength of the crust in northern Tibet?

Line 63-67: "crustal heterogeneities broader plateau to the east", I can see what the authors are trying to say but maybe rewrite those sentences as it does not sounds very clear.

Line 74: Maybe define the two models more clearly in a sentence.

Line 79: Some reference would be nice to justify those choices in composition difference for the orogeny.

Line 83: Why is 3.3 cm/yr used as the reference? Need to justify it somewhere.

Line 86: Actually, the "Asian plate" seems to be referred sometimes to the "upper plate" later, not clear.

Line 95: Repetition from line 91

Line 105: Is the "orogeny" referring to the Himalayas, the plateau or both?

Also not " 's " needed.

Line 105: "rugged", the Tibetan plateau has a relatively homogeneous topography, I can see what the authors meant here but it is not very clear, maybe use another term.

Line 115: Very interesting, should have been developed more I think.

Line 149: Which observed features, not very clear.

Line 169: "Observed" term is not clear what is the reference here.

Line 423: The reference (Barazangi and Ni, 1982) for the question mark is a bit old, a lot of studies have discussed about the lithospheric structure of this area since. This question mark is actually an important issue that the authors should have discussed more in relation to the extension of the plateau to the north.

Line 427: Figure 2 – Worth redefining the model 1 and 2 and also justify the choice of 17.7 Myr.

Line 439: Why is 27 corresponding to partial? What about melt content?

Line 459: Maybe it is worth mentioning that models 1 and 2 are the ones used in the text. Also, there should be some references for the values considered: layer thicknesses, velocities, ...

Line 481: Representing the evolution of the geotherm with time for each collision type would have been very interesting to see and should even be included in the main manuscript, maybe with Fig. 3.

Reviewer #3 (Remarks to the Author)

The authors present new numerical geodynamic models to assess the important role of the Asian crustal rheology on the convergence between Asia and India plates. This is exciting new insight for the geodynamics of a region of significant geological importance and I believe this paper will be of great interest to the readership of Nature communication. I find this paper to be generally well-constructed and well-reasoned. As such I recommend that it is accepted after minor revisions. I do have a few remarks about the manuscript as follows, e.g. L77 indicates line 77 in the text.

1.L77: Although Nature communication has a more general readership, the authors may at least indicate the viscosity ranges of the different crustal rheology in the main article.

2.L115: In the numerical experiment, a stiff Asian lower crust results in the Asian plate underthrusting Indian plate. The best example of this pattern seems to be Pamir to the west as shown in Fig. 1. The interaction between Tarim basin and Tibetan Plateau (and Indian Plate) seems to be a different story as that is often referred as a double-subduction from both sides (e.g. Zhao et al., 2011). The high topography generated (orogenic wedge) for the model is only ~150 km. Apparently this is not the case in Pamir, since the plateau is over 500 km wide (although considerably narrower than the major part of Tibetan plateau). Or do the authors consider this as an "intermediate" state?

3.Continuing with #2, I myself am not quite familiar with Pamir tectonics, so I am sure readers would benefit if the authors explain it a little and list some evidences of a stronger Asian crust in Pamir, if possible.

4.L131: Indeed have different geophysical data suggest a weak lower crust beneath Tibetan Plateau. However, the lower crust is not necessarily weak prior to the convergence, as many studies suggest the weak Tibetan lower crust can be induced by the partial melt/metamorphism after the thickening progress of the crust (e.g. Meissner and Mooney, 1998; Unsworth et al.,

2004). Do the authors claim that lower crust of (proto) Tibet is already weak before the India-Asia collision?

5. Continuing with #3, I am not a modeller and I know little about the technical details. But the interesting results the authors present make me wonder if you can have a stiffer Asian lower crust in the beginning of the convergence procedure but a weaker lower crust due to the thickening process afterwards? In the manuscript the model with a stiffer Asian lower crust results in a subduction of reversed polarization. Will the change of the viscosity affect the following indentation pattern and still produce a high plateau behind the orogeny belt? Is it possible to test the possibility with a numerical thermo-mechanical model?

6.L155: It is fascinating to see the impact of the lower crust rheology on surface velocity and the strain rate. However, the GPS velocity studies (e.g. Gao et al., 2007) have revealed a much larger change of ground motion directions (even like a U-turn towards southeast) and a higher velocity. Could these be reproduced in the numerical experiment with a different parameter (Such as the geometry of the indentation, or with even a weaker Asian lower crust)?

Response to reviewers' comments

Reviewer #1

1. I have found some weaknesses that need to be amended before possible publications. In the introduction, the authors discuss in details tectonic inheritance within the Tibet Plateau, related to Cenozoic deformation for example. And these inherited structures are only simplified in this study by using differences in lower crust rheology. Could you please explain what justify this simplification? Why didn't you test different mantle rheology, or upper crust rheology? Many recent studies indeed argue that tectonic inheritance may imply strong variation in lithosphere mantle strength and therefore rheology.

Response:

We have amended the text to make this choice clearer, and summarized our points.

Indeed, the inherited rheological structures in the lithospheric mantle and upper crust might be as relevant as the lower crust's, although they result in the same outcome: an overall strong/weak lithosphere. All of these heterogeneities are achieved during the previous geological history, and might be found distributed throughout the whole lithosphere thickness. Consequently, heterogeneities in the lower crust are representative of the whole lithosphere's, providing a relevant test for the role of the lithosphere strength.

To support this assumption, we have assessed the lithospheric strength by depth-integrating the stress curves in the Supplementary Fig.2, varying the crustal rheologies, adding the Byerlee law for plasticity and diverse Olivine flow laws (dry-wet). When compared with the available observable inferences (Burov, 2007), models' integrated lithospheric strengths are well comparable with the Earth-like range between 10^{12} and 10^{13} N/m. We have amended the manuscript to refer to readers to the

Supplementary Information.

This test proves that heterogeneities localized in the lower crust provides realistic rheological end-member models and outcomes that are independent of what might be the vertical distribution of heterogeneities in the lithosphere. Any additional heterogeneity, yet in the same geological province, would not change the results.

Furthermore, the lower crust structural variation is well known and documented in the study area. Instead, the upper crust and lithospheric mantle heterogeneities are either negligible or inferred (see next paragraph). Adding inferred and, more importantly, debated lithospheric mantle strength variations would be only speculative.

We digress on the crustal and lithospheric heterogeneities to substantiate our choice. This information is also included in the manuscript, indeed.

Crustal heterogeneities are due to the tectonic inheritance prior to the India-Asia collision and are well documented, as opposed to the unconstrained upper crustal and lithospheric mantle structure, which are poorly known or negligible:

(1) Several geological studies showed that the crust in southern, central and eastern Tibet underwent substantially shortening and may have been as thick as ~50-55 km before the Himalayan orogeny (Murphy et al., 1999; DeCelles et al., 2002; Wallis et al., 2003; Kapp et al., 2005). This implies the existence of strong crust at the proto-southern Asian margin prior to the Cenozoic collision.

Here, the contrast in lithosphere mantle strength is poorly known.

(2) Seismic investigations (Yang et al., 2012; Bao et al., 2015) showed that the lower crust beneath the Tarim is seismically faster than that beneath the Tibet. This implies that a colder and stiffer crust as opposed to a (relatively) warm and weaker.

Similar contrasts in the upper crust and lithosphere mantle exists, but

they appear to be not as strong as the lower crust.

(3) The continents' upper crust is commonly represented by the flow law of Quartzite (Ranalli, 1995), used in our simulations, as a characteristic upper crust, little rheological variations are known. The flow law for the lithosphere mantle is often represented by Olivine's, although results reported in different laboratory studies vary largely (e.g., Burgmann and Dresen, 2008). Assessing the uncertainties in the laboratory-constrained Olivine flow law would go beyond the scope of this paper.

In contrast, the composition for the lower crust has the largest variety among all three lithosphere layers, resulting in extremely diverse flow laws (Burgmann and Dresen, 2008). Here we choose three typical flow laws for the lower crust. 'Dry diabase' represents the strongest one (Mackwell et al., 1998); 'plagioclase An75' represents intermediate (Ranalli, 1995); and 'mafic granulite' represents the weak case (Wang et al., 2012).

2. More important, I did not find in the text the mechanical explanation for such a strong effect of the upper plate lower crust rheology on the style of subduction/collision. Such a mechanical explanation is however necessary if you want to convince that you have find the process that explain the development of the Tibet Plateau.

Response:

Indeed, this is a critical point, which we have thought through. In the submitted version of the paper the mechanical explanation was stated upfront in the abstract, yet not in the text. In the revised version we have added a summary of this explanation in the text, before describing the details of the single models.

To broaden the readability of the paper, we have left in the abstract the explanation in terms of (solid) mechanics, whereas in the text this is recast in terms of orogenic processes and tectonics.

The fundamental reason for a mechanically stronger lithosphere to tend to become the subducting plate is due to its overall larger strength that resists internal deformation. As shown in Figure 2, the lithosphere that eventually becomes the subducting plate experiences less deformation, indicated by the small magnitude and volume of accumulative strain, than the one that becomes the overriding plate.

3. Finally, you aim at modelling, in 3D, the Asia/India system, with effect of tectonic inheritance. A concluding model is to me missing, with the entire indenter (India), delimited in space to the East and West, and with variable (vertically and laterally) rheologies in the upper plate to better discuss the Tibet/Asia collisional system. I am suggesting this final model because it seems that, based on all the modelling material you provide, that you have the technical capacities to run it.

Response:

This is indeed an excellent recommendation. We have included in the paper a new model (now known as figure 3), in which two end-member cases are combined, as recommended by the Reviewer. Indeed, this model improves the paper as the geological implication can be readily understood from the figures.

Reviewer #2

4. The paper shows some interesting results on how crustal rheology has affected the orogen development into a plateau or not. Although the model seems to reconcile quite well with studies from the Pamir region, the results on the formation of the plateau in Tibet appear relatively minor compared to issues related to the growth of the plateau that have been discussed in many studies: extension of the Indian plate to the north, convective removal

of the lithosphere, crustal thickness relation to topography, temperature evolution, the presence of partial melt in the crust, convective thinning of the lithosphere, For instance, the results are not really discussed in comparison with the proposed mechanisms involved in the growth of the Tibetan plateau presented in the introduction. The only discussed issues are mainly the significant tectonic feature of the Himalayan-Tibet orogeny appearing in the models as well as comparison with modelled and present day observed strain rates. Therefore, despite presenting interesting results on the orogeny formation, the paper remains quite vague into really answering the question it poses concerning the growth of the plateau. It is not clear if the authors aim at explaining the initiation of the plateau or its full development. The results remain very interesting in term of describing the initiation of the deformation but it feels it is not enough to answer the question about the formation of the plateau up to its present day state.

Response:

These indications are very helpful and have been accommodated in the revised manuscript clearing further the goal of the paper. As recommended, we have now emphasized that the paper addresses the condition required for the initiation of the plateau and also discussed the crustal heterogeneities in the areas suggested by the Reviewers, i.e. Pamir and northern Tibet. The tectonic structures in these latter are, in fact, compatible with the models predictions, further corroborating our proposed model. Additionally, we have added crustal thickness calculations, which now allow a brief discussion of the relation of crustal thickness to the topography. While the heterogeneities we modeled are inherited from pre-collision tectonics, other such as those indicated by the Reviewer are achieved during the Tibetan growth, and are much less constrained.

We prefer focusing our paper on the most robust observations. Also, we do not address here convective instabilities, as these would impact the

dynamic topography, yet not the thickening of the crust in Tibet, which is the first-order, and more robust, constraint. The computational challenge associated with the high resolution 3-D models of lithospheric-scale structures forces us to confine our models to a shallow depth (i.e., 200 km), we note that deep mantle flow and lithosphere delamination would result in a dynamic component of the topography, which 1) is likely a second-order effect, and 2) has no control on the crustal shortening, which is mostly driven by the convergence, instead.

5. The authors claim in their conclusion at line 174 that “the weak Asian crust can explain the formation of the plateau behind an orogeny wedge and the observed undeplating Indian lithosphere”. However, from their results it is difficult to see how the actual observations from the state of the plateau today can be related to the models.

Response:

This is highlighted in Figure 1, which summarises the relevant observations used to validate our models. We have amended the text to emphasise the comparison, reinforcing the concept that if the upper plate is stronger than the lower plate the first-order observation is no underplating and no plateau formation. We reiterate that our modeled Plateau is not as wide as that observed, due to the computational challenge to capture a larger geographic region, and that the results are comparable conceptually to the formation of the Tibetan Plateau.

6. The paper would be a bit more clear maybe if Figure 2 would be replaced by Figure 3 to present the initiation and stages of the deformation first and then show on Figure 3 the final stage of the simulation to discuss how it compares with current topography and geophysical observations from Tibet and Pamir.

Response:

We have rearranged the figures, to accommodate the new model the Reviewer recommended. We agree that the original figure 3 is likely not serving as is and have replaced it with the crustal thickening (now known as figure 4), as suggested by the Reviewers. This information is clearer to a wider audience, and allows the immediate comparisons with geophysical measurements of crustal thickness in Tibet, while embedding the concept of isostatic topography.

7. The topography should be compared with real topo data to discuss the differences, for instance, from the Figure 3 the plateau generated from model 1 seems to be only 200 km wide, how does this compare with real topography data which clearly reflects a wider plateau.

Response:

We have amended the text and replaced the figure 3 with a crustal thickness map (now known as figure 4), as recommended by the Reviewers. This presents a more robust case for the comparisons with the distribution of crustal thickness in the study area. Variations to the topography might be, in fact, affected by many other factors not included in our modelling, although these are minor.

Regarding the width of the plateau, this is, of course time-dependent. In fact, original figure 3 shows the topography growth after ~580 km of convergence. At this model time, the widest plateau is ~500-600 km, thus showing that the width of the plateau is almost linearly increasing with convergence, as can be easily understood. One can expect that the plateau will become wider as convergence proceeds in the weak Asian lithosphere model, therefore for a longer model time, totaling up to the ~2500 km of India's northward motion since collision, the width of the plateau would be comparable. Because we use a high-resolution grid in the modeling, it is computationally difficult to enlarge the model size to the real situation. As stated before, we agree that figure 3 does not serve

the comparison as is and have removed it.

8. Also how does the thickness of the crust and temperature evolution through the models compare with the studies from the plateau?

Response:

Indeed, this piece of information is relevant to the audience. We have thus replaced the original figure 3 with the crustal thickness field for the models, added model isotherms to the composition field, in figure 2 and 3. This allows for comparisons with the recent estimates for the present-day crustal thickness and thermal structure across the Himalaya orogen and Tibet.

9. The 3D code used is described as “Fully thermodynamically coupled and accounts for melting processes”. A wide variety of studies have agreed on the presence of partial melt in the middle/lower crust of the plateau that would affect strongly the rheology of the plateau. From Figure 2, it seems that melt appears on model 2 but not model 1. It is worth discussing this point, why melt is not needed, how do both crustal thickness and geotherm evolve, how does this affect melt generation or not, and therefore the rheology. Maybe a closer focus on the crust in a picture could also help.

Response:

This is a point raised (and addressed) above. We agree that crustal thickening and melting are indeed points to further discuss. We have added this information (see above) and briefly discussed how the models compare to the observation available.

10. It is in some way paradoxical to discuss the heterogeneity of the whole orogen by comparing two “homogeneous” models with different rheology. It is worth to justify that choice. The authors use a 3D modelling so maybe it makes more sense to eventually include the crustal heterogeneities directly

into one single model if computationally possible. The authors discuss the heterogeneities between Pamir and Tibet but they should also probably emphasize on the heterogeneities inside the plateau between west, central and eastern Tibet.

Response:

As recommended by the Reviewer, we have added a third model where the lateral heterogeneities are embedded in the single model. The revised text now discusses this additional model. We are confident that this answers thoroughly to this request. We also note that the paper does address diverse heterogeneities in the area, including east and west Tibet. Although a model capturing all these complexities is, in principle, possible, we note that, in fact, the heterogeneities are of two types (relatively stronger/weaker), so that we find the model embedding these two is illustrative enough.

11.line 122 to 135, It is interesting to compare the model features with observed structures but those are very difficult to see on the Figures. The authors need to find a way to describe it on the models better. This comment goes back to a previous comment on having a picture of the model more focused on the crust.

Response:

In the current revised manuscript we have preferred to leave the comparison between models and observations to the first-order features of lithospheric geometry, subduction and thrusting polarity, as these are dramatically different in the end-members models. These features are in agreement with deep geometries, i.e. underplating/overriding, and large-scale structures, i.e. underthrusting from north and south beneath Tibet, and provide an instantaneous assessment of the models.

Yet, as requested, we have added detailed figures in the Supplementary Information to allow some more detailed comparisons.

Furthermore, the issue of how the shortening is accommodated within the plateau might be misleading: Whether this happens through discrete thrusts planes or by pure shear, i.e. homogeneous thickening, is still a matter of debate, yet it would not change the first-order observation on the ability of the crust to propagate the stress, which depends on viscosity, not plasticity.

12. line 144 to 150, it is actually a very important point that I think the authors should have explored further particularly in the sense of trying to understand how the plateau extended to the north, by discussing the different mechanisms involved in the process as well as the rheological differences between west, central and east Tibet.

Response:

We have expanded this paragraph and explored further the comparison with the models. This is now done integrating the observation in Fig.1 with the models outcomes in Fig.2, 3 and 5.

13. On the GPS comparison, the authors tend to repeat themselves and should maybe instead go more in detail in the comparison between synthetic and observed values of strain rates.

Response:

We have shortened the discussion of the velocity field, and enlarged the discussion of the strain rates. This part now includes a comparison with the current deformation in the area, as well as some inferences on the deep structure of Tibet.

14. On Figure 4, are the arrows velocities? Should be clarified.

Response:

Yes, the arrows represent velocities at surface. We have clarified in the figure caption.

15. The results are compared with present day GPS measurements but what time of the simulation is represented on Figure 4. How does it compare then with real data? This section is relatively long and therefore looks like a major part of the manuscript. The authors should find a better way to compare those in the text and in the pictures.

Response:

We have rewritten this part of the manuscript to illustrate first, the deformation compatibility with the observation, that is GPS and strain rates in Tibet, and use this to support inferences on the deep structure.

16. Line 40: On the descriptions of the models, the authors should also mention the effects of convective thinning of the lithosphere to the north which has been proposed as another mechanism to explain the extension of the high topography further north (See for example Jimenez-Munt 2008).

Response:

We have rewritten the manuscript to focus more on the crustal thickening. The topography, in fact, is the result of crustal thickening in the first place, plus the effect of deep mantle dynamics. Focusing on the crustal thickness, and less on the topography, we mean to address the static component of the topography, which is well established, while we leave out the discussion of mantle dynamic component, which remains quite debated. We have added the references suggested pointing out another mechanism.

17. Line 44: Not only seismic but a variety of other studies such as magnetotellurics have contributed to study the deformation of the plateau.

Response:

Several MT studies, such as Wei et al. (2001) and Unsworth et al. (2005), have been cited to further illustrate the complex crustal/lithospheric

structure beneath the Tibetan Plateau.

18.Line 47: More recent references than Barazangi and Ni, 1982 have been published.

Response:

More recent relevant references, such as Li et al. (2008), Agius & Lebedev (2013) and Bao et al. (2015), have been included.

19.Line 49: In addition to Zhao et al. 2010, the authors could also add Agius and Lebedev, 2013 which also highlights quite well the heterogeneities of the lithospheric structure of the plateau and particularly discuss its effects on thermal anomalies. The later could be a very good addition to the temperature data extracted from the 3D simulations.

Response:

Indeed, the paper by Agius & Lebedev (2013) is very relevant to the presented study. We have added it.

20.Line 52: (Fig. 1b) instead of (Fig. 1d)?

Response:

We have corrected this error.

21.Line 51-54: No mentioning of Fig. 1c, maybe some short sentence would be nice.

Response: Amended

22.Line 54: It should be discussed what is defined as an interrogation mark on the Fig. 1d.

Response:

The interrogation mark indicates a special lithospheric region sandwiched between Indian and Asian plates, whose affinity is unclear.

This explanation has been included in the figure caption.

23. Line 60: Maybe Yin and Harrison, 2001 could be a good reference here also as it is a review of the geology of Tibet.

Response:

Thanks. The citation of Yin & Harrison (2000) has been added.

24. Line 63: What about the strength of the crust in northern Tibet?

Response:

We have amended the text to discuss this aspect too. In fact, seismic analyses infer a - relatively - stronger crust in the whole area from north Tibet to Pamir-Tien Shan. While our modeling mainly focuses on formation of the Plateau following collision, where strength contrasts between the Tarim Basin and the proto-southern Asian margin played a relevant role, we have now highlighted that similar contrasts might be relevant during the evolution of the Tibetan Plateau, and discussed the inversion of the tectonics in the Pamir and north Tibet as a potential example.

The present-day southern and central Tibet represents the main body of the pre-collision southern Asia margin. It was estimated that the Lhasa and Qiangtang terranes were shortened >470 km over a present-day distance of 473 km (>50%) during Late Cretaceous-Early Tertiary time (Kapp et al., 2005). This suggests a thick and soft crust in southern and central Tibet prior to the India-Asia collision, which is the focus of the modelling.

In contrast, the thickness and strength of the northern Tibetan crust prior to the collision is poorly known. U-Pb dating of zircons in Barrovian-facies metamorphic rocks at the eastern margin of the Songpan-Ganzi terrane gives ages of c. 65 Ma (Wallis et al., 2003). This suggests the crust there was already thick before the India-Asia

collision.

25.Line 63-67: “crustal heterogeneities broader plateau to the east”, I can see what the authors are trying to say but maybe rewrite those sentences as it does not sounds very clear.

Response: This paragraph has been rewritten.

26.Line 74: Maybe define the two models more clearly in a sentence.

Response:

We have amended this paragraph to clearly indicate the model characteristics. We have also added a third model, with both heterogeneities, as recommended by the Reviewers.

27.Line 79: Some reference would be nice to justify those choices in composition difference for the orogeny.

Response:

Two relevant references, such as Mackwell et al. (1998) and Wang et al. (2012) have been added.

28.Line 83: Why is 3.3 cm/yr used as the reference? Need to justify it somewhere.

Response:

A convergence rate of 3.3 cm/yr is in agreement of the estimate of 3.5 ± 1.3 cm/yr for Cenozoic Asian shortening rate by Guillot et al. (2003). This statement has been added in the text to explain the reason.

29.Line 86: Actually, the “Asian plate” seems to be referred sometimes to the “upper plate” later, not clear.

Response:

We have clarified this by rephrasing Line 86 as follows:

In the rest of the paper we refer to the incoming plate as the Indian plate or indenter, and the broader retro-continent as the Asian plate or upper plate.

30. Line 95: Repetition from line 91

Response: We have rephrased the sentence to avoid repetition.

31. Line 105: Is the “orogeny” referring to the Himalayas, the plateau or both?

Also not “ ’s ” needed.

Response:

This has been clarified as follows: “The plateau then grows in the upper plate interiors to a height similar to the Himalayan-Tibetan orogenic system.”

32. Line 105: “rugged”, the Tibetan plateau has a relatively homogeneous topography, I can see what the authors meant here but it is not very clear, maybe use another term.

Response: We have amended the text, accordingly.

33. Line 115: Very interesting, should have been developed more I think.

Response:

Indeed, we are planning to explore this aspect further in future work. However, this case does not really compare to the Tibetan Plateau, so that length constraints prevent us to explore further here.

34. Line 149: Which observed features, not very clear.

Response:

The observed features indicate the east-west variations in the crustal and lithospheric structures along the northern Tibetan margin. We have amended the text and spelled out the features observed.

35. Line 169: "Observed" term is not clear what is the reference here.

Response:

The observed here referred to Guillot et al. (2003), which has been added as a citation.

36. Line 423: The reference (Barazangi and Ni, 1982) for the question mark is a bit old, a lot of studies have discussed about the lithospheric structure of this area since. This question mark is actually an important issue that the authors should have discussed more in relation to the extension of the plateau to the north.

Response:

We included a few recent references, such as Zhao et al. (2011) for the lithospheric structure along the profile CC'. We note that the paper focuses on the early formation of the Tibetan Plateau and first-order features. Then, by the same token we speculate that similar heterogeneities controlled the overall evolution of the Plateau, including its northernmost expansion, and provide references. Yet, this part remains too speculative, and we prefer to leave it as a minor point of the paper.

37. Line 427: Figure 2 – Worth redefining the model 1 and 2 and also justify the choice of 17.7 Myr.

Response:

Good suggestions. We have redefined 'Model-1' as 'weaker Asian crust' and 'Model-2' as 'stiffer Asian crust' in the caption. The snapshots taken at 17.7 Myr is taken during steady-state continental collision, and well represents two distinctive deformation styles due to contrast in crustal rheology. At a later stage, the topography front approaches the back wall of the models, where free slip boundary condition is applied, and the

modeled results will become unrealistic.

38.Line 439: Why is 27 corresponding to partial? What about melt content?

Response:

In 3-D code I3EVIS, composition 27 corresponds to partial melting counterpart of composition 7 (i.e., continental upper crust). Although the code allows modelling the melt content as a function pressure and temperature above the granite solidus, this is not addressed here. The thickening and formation of the plateau are direct consequences of convergence, while the mass is conserved. Therefore, whether the crust is under condition of partial melting or not, does not affect the crustal mass and the finite strain (thickness)

39.Line 459: Maybe it is worth mentioning that models 1 and 2 are the ones used in the text. Also, there should be some references for the values considered: layer thicknesses, velocities, ...

Response:

We have mentioned that Model 1 and 2 are the ones used in the comment column. Convergence rates are mainly based on Guillot et al. (2003). Lithospheric layering for indenter is based on the result by Singh et al. (2015); whereas lithospheric layering for upper plate is referenced from Hacker et al. (2015). The above information has been included in Supplementary Table 1.

40.Line 481: Representing the evolution of the geotherm with time for each collision type would have been very interesting to see and should even be included in the main manuscript, maybe with Fig. 3.

Response:

Good suggestion. We have overlapped the isotherms onto the composition section in Figure 2 and 3. Unfortunately, for length

restriction, we will not be able to discuss further. However, we note that the temperature evolution and potential melting remain second-order features, so well worth investigating in further research.

Reviewer #3

41.L77: Although Nature communication has a more general readership, the authors may at least indicate the viscosity ranges of the different crustal rheology in the main article.

Response: We have clarified the viscosity contrast used.

42.L115: In the numerical experiment, a stiff Asian lower crust results in the Asian plate underthrusting Indian plate. The best example of this pattern seems to be Pamir to the west as shown in Fig. 1. The interaction between Tarim basin and Tibetan Plateau (and Indian Plate) seems to be a different story as that is often referred as a double-subduction from both sides (e.g. Zhao et al., 2011). The high topography generated (orogenic wedge) for the model is only ~150 km. Apparently this is not the case in Pamir, since the plateau is over 500 km wide (although considerably narrower than the major part of Tibetan plateau). Or do the authors consider this as an “intermediate” state?

Response:

We have rewritten the text illustrating this case, as correctly suggested, as a relevant one. The evolution of Pamir is, in fact, very compatible with our modelling outcomes: Cenozoic underthrusting, thickening and plateau propagation to the north in the Pamir are truncated by the indentation onto the current stiffer crust in north Pamir-Tien Shan, forcing the polarity inversion.

The lateral extent of Pamir, and the plateaux in general, likely depends on the protracted indentation which in nature is ~4 times of that used in our

models, that is ~2500 km as opposed to ~580 km; this likely explains the width of 150 as opposed to 500 km. Due to numerical difficulty, our models could not simulate a wide enough box as the real case.

43. Continuing with #2, I myself am not quite familiar with Pamir tectonics, so I am sure readers would benefit if the authors explain it a little and list some evidences of a stronger Asian crust in Pamir, if possible.

Response:

We have added the two references supporting this evidence and have rewritten the whole manuscript to illustrate better the case of Pamir.

44. .L131: Indeed have different geophysical data suggest a weak lower crust beneath Tibetan Plateau. However, the lower crust is not necessarily weak prior to the convergence, as many studies suggest the weak Tibetan lower crust can be induced by the partial melt/metamorphism after the thickening progress of the crust (e.g. Meissner and Mooney, 1998; Unsworth et al., 2004). Do the authors claim that lower crust of (proto) Tibet is already weak before the India-Asia collision?

Response:

The focus of this paper is on the relative strength of the Indian and Asian crusts. This does support inferences on the relative strength in the studied area, whereas it does not allow inferences on the absolute strength of the Asian crust. We have cleared this in the text.

45. Continuing with #3, I am not a modeller and I know little about the technical details. But the interesting results the authors present make me wonder if you can have a stiffer Asian lower crust in the beginning of the convergence procedure but a weaker lower crust due to the thickening process afterwards? In the manuscript the model with a stiffer Asian lower crust results in a subduction of reversed polarization. Will the change of the

viscosity affect the following indentation pattern and still produce a high plateau behind the orogeny belt? Is it possible to test the possibility with a numerical thermo-mechanical model?

Response: While this is possible to test with the numerical models we can reach a conclusion following some reasoning. For a constant viscosity crust/lithosphere viscosity during convergence, that is thickening is faster than thermal diffusion, the thickness increase results in a linear increase in tensile strength. Hence, a thicker crust is increasingly stiffer, thus propagating further the stress (see Molnar and Lyon-Caen, 1988). This can only be reversed if some thermal process, other than diffusion, occurs, such as delamination. This does not alter the results found here, but would only affect the evolution of the Tibetan Plateau. While interesting, this test remains out of the scope of this paper.

46.L155: It is fascinating to see the impact of the lower crust rheology on surface velocity and the strain rate. However, the GPS velocity studies (e.g. Gao et al., 2007) have revealed a much larger change of ground motion directions (even like a U-turn towards southeast) and a higher velocity. Could these be reproduced in the numerical experiment with a different parameter (Such as the geometry of the indentation, or with even a weaker Asian lower crust)?

Response: While this is an interesting aspect, it is likely beyond the scope of this paper. In fact, the large-scale motion pattern is likely best addressed in models reproducing self-consistent subduction along the Indian and Southeast Asian margin. This was the topic of investigation of specific models published by some of the co-Authors, e.g. Li et al., 2013, Duretz et al., 2014, Capitanio et al., 2016.

Reviewers' Comments:

Reviewer #2:

Remarks to the Author:

The manuscript has been strongly improved, reads very well and illustrates the findings more clearly. The introduction of the model 3 particularly synthesizes the message of the paper very nicely as for a broader audience it can be directly related to the topography map of the Himalayan-Tibetan orogeny on figure 1.

I also really appreciate the effort of the authors to take into account and reply clearly to each comments.

Just two very minor comments

Lines 65 to 68: Maybe you can reformulate or remove the "therefore" which is kind of confusing in the order of the sentences.

line 195: " The surface velocities predicted" End of sentence missing?

Reviewer #3:

Remarks to the Author:

I have no further comments to the revised manuscript, as the authors have rewritten the manuscript and added new materials to address my previous comments.